

# c-Kit modifies the inflammatory status of smooth muscle cells

Lei Song[1], Laisel Martinez[2], Zachary M. Zigmond[1], Diana R. Hernandez[2], Roberta M. Lassance-Soares[2], Guillermo Selman[2] and Roberto I. Vazquez-Padron[2]

[1] Department of Molecular and Cellular Pharmacology, Leonard M. Miller School of Medicine, University of Miami, Miami, FL, United States of America
[2] DeWitt Daughtry Family Department of Surgery, Division of Vascular Surgery, Leonard M. Miller School of Medicine, University of Miami, Miami, FL, United States of America

## ABSTRACT

**Background.** c-Kit is a receptor tyrosine kinase present in multiple cell types, including vascular smooth muscle cells (SMC). However, little is known about how c-Kit influences SMC biology and vascular pathogenesis.

**Methods.** High-throughput microarray assays and *in silico* pathway analysis were used to identify differentially expressed genes between primary c-Kit deficient ($Kit^{W/W-v}$) and control ($Kit^{+/+}$) SMC. Quantitative real-time RT-PCR and functional assays further confirmed the differences in gene expression and pro-inflammatory pathway regulation between both SMC populations.

**Results.** The microarray analysis revealed elevated NF-κB gene expression secondary to the loss of c-Kit that affects both the canonical and alternative NF-κB pathways. Upon stimulation with an oxidized phospholipid as pro-inflammatory agent, c-Kit deficient SMC displayed enhanced NF-κB transcriptional activity, higher phosphorylated/total p65 ratio, and increased protein expression of NF-κB regulated pro-inflammatory mediators with respect to cells from control mice. The pro-inflammatory phenotype of mutant cells was ameliorated after restoring c-Kit activity using lentiviral transduction. Functional assays further demonstrated that c-Kit suppresses NF-κB activity in SMC in a TGFβ-activated kinase 1 (TAK1) and Nemo-like kinase (NLK) dependent manner.

**Discussion.** Our study suggests a novel mechanism by which c-Kit suppresses NF-κB regulated pathways in SMC to prevent their pro-inflammatory transformation.

Corresponding author
Roberto I. Vazquez-Padron,
rvazquez@med.miami.edu

# INTRODUCTION

The c-Kit receptor tyrosine kinase is a proto-oncogene and stem cell marker that has been recently implicated in vascular pathogenesis. Widely recognized for its proliferative and anti-apoptotic role in hematopoietic stem and progenitor cells (*Bernstein et al., 1991*), c-Kit signaling is now known to increase endothelial permeability (*Kim et al., 2014*; *Im, Song & Suh, 2016*), and regulate the phenotype of smooth muscle cells (SMC) in the vasculature (*Wang et al., 2007*; *Davis et al., 2009*). On one hand, animal models indicate that c-Kit activation by its ligand the stem cell factor (SCF) plays an important role in the development of both arterial and venous intimal hyperplasia (IH) (*Hollenbeck et al., 2004*; *Wang et al., 2006*; *Wang et al., 2007*; *Skartsis et al., 2014*). On the other hand, c-Kit

expression preserves the SMC contractile phenotype (*Davis et al., 2009*), and protects arteries from excessive atherosclerosis (*Song et al., 2016b*). Therefore, whether c-Kit is beneficial or detrimental for the vasculature is still a matter of debate and warrants further investigations.

Expression of c-Kit and SCF in vascular myofibroblasts and SMC seems to be tightly regulated by a variety of physiological and pathological triggers. c-Kit positive SMC populate the intima of arteries and veins after vascular injury in models of angioplasty and vein grafting (*Hollenbeck et al., 2004*; *Wang et al., 2006*; *Wang et al., 2007*). Temporal up-regulation of both c-Kit and SCF are reported after injury in these models (*Hollenbeck et al., 2004*; *Wang et al., 2006*; *Wang et al., 2007*), where c-Kit dependent induction of the Akt-Bcl-2 cascade is thought to mediate the anti-apoptotic and migratory SMC phenotype responsible for IH (*Wang et al., 2007*). The formation of venous IH in arteriovenous fistulas (AVF) also occurs secondary to the activation of c-Kit expressing adventitial progenitors and migration of c-Kit positive myofibroblasts to the intima (*Skartsis et al., 2014*). Both animal and human AVF demonstrate higher numbers of c-Kit expressing SMC after surgery compared to preoperative veins, along with a transitional increase in SCF levels in animal models after AVF creation (*Skartsis et al., 2014*).

A small population of c-Kit expressing myofibroblasts/SMC was also found in the pulmonary arteries of patients with idiopathic pulmonary arterial hypertension but not in healthy controls (*Montani et al., 2011*). In line with this evidence, both SCF and c-Kit were up-regulated in pulmonary arterioles of experimental animals with pulmonary hypertension, where c-Kit colocalized with cells in the endothelium, media and adventitia (*Young et al., 2016*). In the latter model, SCF/c-Kit signaling promotes pathological remodeling and pulmonary vascular cell proliferation after hypoxic stimulation via activation of the ERK1/2 pathway (*Young et al., 2016*). Paradoxically, the presence of c-Kit in human primary pulmonary artery SMC up-regulates the transcription factor myocardin and preserves the contractile SMC phenotype (*Davis et al., 2009*), suggesting a protective role of c-Kit under certain vascular conditions.

Along with the up-regulation of c-Kit, the above evidence indicates that SCF is expressed and released in the vasculature in response to different insults (*Miyamoto et al., 1997*; *Hollenbeck et al., 2004*; *Wang et al., 2007*). Vascular SCF exists in both membrane-bound and soluble forms, thus its ability to promote cell recruitment and elicit autocrine and paracrine responses, as well as cell-to-cell stimulation (*Hollenbeck et al., 2004*; *Lennartsson & Ronnstrand, 2012*; *Skartsis et al., 2014*). The soluble form of SCF is generated by alternative splicing or released through the proteolytic action of matrix metallopeptidase 9 (MMP-9) (*Hollenbeck et al., 2004*; *Bengatta et al., 2009*; *Lennartsson & Ronnstrand, 2012*; *Klein, Schmal & Aicher, 2015*). This latter enzyme is also up-regulated in vascular remodeling processes, thereby perpetuating the local effects of the SCF/c-Kit pathway (*Hollenbeck et al., 2004*; *Skartsis et al., 2014*). Interestingly, the soluble and membrane-bound SCF isoforms seem to have different effects on c-Kit activation (*Miyazawa et al., 1995*). The former causes rapid and transient stimulation and autophosphorylation of the receptor as well as fast degradation, whereas the latter leads to sustained activation (*Miyazawa et al., 1995*). This observation suggests that cell-to-cell interactions between

SCF and c-Kit expressing SMC have the potential to significantly modify their respective phenotypes and nearby microenvironment. The concomitant expression of SCF and c-Kit in various cell types (*Lennartsson & Ronnstrand, 2006*; *Zakiryanova et al., 2014*), including myofibroblasts/SMC (*Hollenbeck et al., 2004*; *Wang et al., 2007*; *Skartsis et al., 2014*), also implies the presence of an autocrine loop for the activation of this receptor. Unfortunately, the available data on the role of c-Kit in vascular remodeling processes is still scarce, and more information is particularly needed on the c-Kit mediated pathways that regulate the SMC phenotypic transformation (*Wang et al., 2007*).

In this work, we used high-throughput microarray analyses to identify differentially expressed genes as a result of c-Kit loss of function in arterial SMC isolated from mutant and littermate control mice. We combined *in silico* pathway analyses and confirmatory assays to further investigate the gene expression profiles of stimulated SMC under both experimental conditions. We showed increased NF-κB activation in c-Kit deficient SMC compared to their wild type counterparts. Furthermore, we demonstrated that these changes were associated with a heightened state of vascular inflammation, as indicated by the elevated protein expression of pro-inflammatory mediators in c-Kit deficient SMC. Outcomes from this study challenge the existing belief that vascular c-Kit expression is pathological, and suggest instead a beneficial contribution of this signaling axis for the preservation of SMC's anti-inflammatory status under adverse conditions.

## MATERIALS AND METHODS

### Smooth muscle cell isolation and culture

Primary aortic SMC were isolated from c-Kit deficient ($Kit^{W/W-v}$) mice and control littermate mice ($Kit^{+/+}$) (Stock #100410, The Jackson Laboratories, Bar Harbor, ME, USA) (*Bernstein et al., 1990*) using the explant technique (*Metz, Patterson & Wilson, 2012*) with minor modifications. Briefly, mouse aortas were digested with collagenase type II (5 mg/mL, Worthington, Lakewood, NJ, USA) at 37 °C for 1 h, after which they were transferred to 10% FBS and cut with a scalpel into small pieces. Individual SMC migrated out of the explants within 1 week of culture. Cells were maintained in DMEM-F12-FBS (5:3:2; Thermo Fisher Scientific, Waltham, MA) supplemented with 100 I.U./ml penicillin, 100 μg/ml streptomycin, 2 mM L-glutamine, 1 mM sodium pyruvate, and 0.075% sodium bicarbonate (*Metz, Patterson & Wilson, 2012*). Primary cells were maintained at ∼90% confluency and used within three passages to avoid fibroblast-like phenotypic switching. All animal procedures were performed according to the National Institutes of Health guidelines (Guide for the Care and Use of Laboratory Animals) and approved by the University of Miami Miller School of Medicine Institutional Animal Care and Use Committee (protocol 15-114).

### RNA microarray and pathway analysis

Total RNA was isolated from $Kit^{+/+}$ and $Kit^{W/W-v}$ SMC using the Quick-RNA MiniPrep kit (Zymo Research, Irvine, CA, USA). RNA quality was validated in the Agilent 2100 Bioanalyzer (Agilent Technologies, Santa Clara, CA, USA) before being sent to Ocean Ridge Biosciences (Palm Beach Gardens, FL, USA) for Mouse MI-Ready Gene Expression Microarray analysis. Once in Ocean Ridge Biosciences, RNA processing included a

30-minute digestion with RNase-free DNase I (Epicentre, Madison, WI, USA) at 37 °C followed by purification using the AgenCourt RNAClean XP bead method (Beckman Coulter, Indianapolis, IN, USA). Biotin-labeled complementary RNA (cRNA) was prepared from 2 μg per sample of re-purified RNA by the method of *Van Gelder et al. (2006)* (Van Gelder's Multi-gene expression profile—US Patent 7049102). Eighteen micrograms of biotinylated cRNA per sample were fragmented, diluted in formamide-containing hybridization buffer, and loaded onto the surface of the Mouse MI-Ready microarray slides enclosed in custom hybridization chambers. The slides were hybridized for 16–18 h under constant rotation in a Model 400 hybridization oven (Scigene, Sunnyvale, CA, USA). After hybridization, the microarray slides were washed under stringent conditions, stained with Streptavidin-Alexa-647 (Life Technologies, Waltham, MA, USA), and scanned using an Axon GenePix 4000B scanner (Molecular Devices, Sunnyvale, CA, USA). Probe intensities were calculated for each feature on each microarray by subtracting the median local background from the median local foreground for each probe. Data for all manufacturer-flagged probes and visually flagged probes impacting >25% of samples were removed. Data for visually flagged probes impacting <25% of samples were replaced with the sample average for the probe. Probe intensities were transformed by calculating the base 2 logarithm of each value. Array-specific detection thresholds (T) were calculated by adding three times the standard deviation of the median local background and the mean negative control probe signal. Probe intensity and T were normalized by subtracting the 70th percentile of the mouse probe intensities and adding back the mean of the 70th percentile across all samples as a scaling factor. The data were filtered to select for mouse probes showing signal above the normalized T in at least 25% of the samples; data for control sequences and other non-mouse probes were removed. Mouse probe sequences were annotated using a BLAST analysis of the Ensembl Mouse cDNA database version 84 (EMBI-EBI, Cambridge, UK). Gene expression differences between $Kit^{+/+}$ and $Kit^{W/W-v}$ SMC were considered statistically significant if $p < 0.05$ by *t-test*.

For pathway analysis, genes with statistically significant expression differences in microarray analysis were imported into the Ingenuity Pathway Analysis software (http://www.ingenuity.com/; Ingenuity Systems, Redwood City, CA, USA). The Core Analysis was used to identify the canonical pathways associated with the differentially expressed genes. Pathway overlap and *p*-value calculations were performed using the reference gene set in the Ingenuity Knowledge Base, where only molecular relationships (direct and indirect) that have been experimentally observed were considered. The Molecule Activity Predictor tool was used to estimate activation or inhibition of pathway branches based on the observed gene expression fold changes in $Kit^{W/W-v}$ vs. $Kit^{+/+}$ SMC.

## Quantitative real-time RT-PCR

Relative gene expression of selected mRNA transcripts was evaluated using TaqMan Gene Expression Assays (Applied Biosystems, Foster City, CA, USA). Total RNA was isolated as described above, and cDNA synthesized with the High-Capacity cDNA Reverse Transcription kit (Applied Biosystems, Foster City, CA, USA). Real-time RT-PCR was performed on an ABI Prism 7500 Fast Real-Time PCR System (96-well plate) (Applied Biosystems, Foster

City, CA, USA) using primers/probe sets complementary to the genes of interest (*ActB*, Mm00607939_m1; *Ccl2*, Mm00441242_m1; *Ikbka*, Mm00432529_m1; *Ikbkb*, Mm01222247_m1; *Ikbkg*, Mm00494927_m1; *Il6*, Mm00446190_m1; *Kit*, Mm00445212_m1; *Kitl*, Mm00442972_m1; *Map3k14*, Mm0048444166_m1; *Mmp2*, Mm00439498_m1; *Mmp9*, Mm00442991_m1; *Nfkb2*, Mm00479807_m1; *Nfkbia*, Mm00477798_m1; *Nos2*, Mm00440502_m1; *Ptgs2*, Mm00478374_m1; *RelB*, Mm00485664-m1; *Tnf*, Mm00443258_m1). Relative gene expression was determined using the $\Delta\Delta$CT method (*Livak & Schmittgen, 2001*) and normalized with respect to *ActB*.

## Gene rescue and knockdown

Gene rescue in c-Kit deficient (Kit$^{W/W-v}$) SMC was performed using a lentiviral vector (pRVPG24). This rescue vector was constructed by inserting a blunted BsrBI-NotI digested DNA fragment (3.6 Kb), containing the coding region of the mouse Kit cDNA under the murine phosphoglycerate kinase (PGK) promoter, into the blunted EcoRV-ClaI digested pLenti CMV PuroDest vector (Addgene Inc., Cambridge, MA, USA). Third generation lentiviral stocks were produced in HEK-293 cells co-transfected with the lentiviral vector and the packaging and envelope plasmids psPAX2 and pMD2.G (Addgene Inc.). Transfections were done with the jetPRIME transfection kit (Polyplus, New York, NY). Infected cells (100 MOI) were selected in DMEM-F12-FBS (5:3:2; Thermo Fisher Scientific, Waltham, MA, USA) supplemented with 100 I.U./ml penicillin, 100 µg/ml streptomycin, 2 mM L-glutamine, 1 mM sodium pyruvate, 0.075% sodium bicarbonate, and 10 µg/ml puromycin (Sigma, St Louis, MO).

Knockdown of TAK1 or NLK in c-Kit wild type (Kit$^{+/+}$) SMC was performed using pooled lentiviral particles carrying different target siRNAs (Table S1; Applied Biological Materials, Richmond, Canada). An anti-GFP siRNA was used as control. Transduced cells were puromycin-selected as described above. All gene modifications were confirmed by analytical flow cytometry or Western blot (WB).

## Flow cytometry analysis

c-Kit surface expression was evaluated by flow cytometry in SMC stained with an anti-CD117 antibody (CD117-PE, Cat# 130-091730, Miltenyi Biotec, San Diego, CA, USA). Analytical flow cytometry was performed on a BD FACS Canto II (BD Biosciences, San Jose, CA, USA) using the BD FACSDiva software (Becton Dickinson, Franklin Lakes, NJ, USA). Data were analyzed using the FlowJo software (Ashland, OR, USA).

## Western blot and immunoprecipitation (IP)

Whole cell protein lysates were prepared in RIPA buffer supplemented with 200 mM phenylmethylsulfonyl fluoride (PMSF), 100 mM sodium orthovanadate (Santa Cruz Biotechnology, Dallas, TX, USA), and a complete protease inhibitor cocktail (Roche Life Science, Indianapolis, IN, USA). Lysate concentration was determined using a commercial Bradford's protein assay kit (BioRad, Hercules, CA, USA). For WB analysis, ~50 µg of sample was loaded into a NuPAGE 4–12% Bis-Tris SDS-polyacrylamide gel (Thermo Fisher Scientific, Waltham, MA, USA) and subsequently transferred to a PVDF membrane (GE Healthcare, Marlborough, MA, USA). Specific proteins were detected using antibodies

against c-Kit (1:1,000, Cat# sc-1494; Santa Cruz Biotechnology, Dallas, TX, USA), MCP-1, MMP-2, TAK1 (1:500, Cat# sc-1785, sc-1839, and sc-6838, Santa Cruz Biotechnology, Dallas, TX, USA), NLK (1:1,000, Cat# ab26050; Abcam, San Francisco, CA, USA), Src (1:500, Cat# 2108S; Cell Signaling Technology, Danvers, MA, USA), and β-Actin (1:5,000, Cat# A5316; Sigma, St. Louis, MO, USA). Bound antibodies were detected after sequentially incubating the membranes with HRP-conjugated secondary antibodies. The Amersham ECL Western Blotting Detection Reagent (GE Healthcare, Marlborough, MA, USA) or SuperSignal West Femto Maximum Sensitivity Substrate Reagent (Thermo Fisher Scientific, Waltham, MA, USA) were used for signal detection. Images were analyzed using ImageJ Pro 5.0.

For co-IP, ∼200 µg of protein lysate was incubated at 4 °C for 4 h with 1 µg of anti-c-Kit (Cat# A4502, Dako, Santa Clara, CA, USA) or TAK1 antibodies and 20 µl of Protein A/G PLUS-Agarose microbeads (Santa Cruz Biotechnologies, Dallas, TX, USA). Microbeads were washed with cold RIPA buffer before WB analysis for c-Kit, TAK1, or NLK as indicated above.

## NF-κB promoter activity

Primary SMC were transfected with a commercial mix of NF-κB Luc-reporter plasmids (Qiagen, Germantown, MD, USA) using the Axama Basic Nucleofector Primary Smooth Muscle Cells electroporation kit (Cat# VPI-1004, Lonza, Walkersville, MD, USA). Transfected cells were incubated with 1-palmitoyl-2-(5-oxovaleroyl)-*sn*-glycero-3-phosphocholine (POVPC; Avanti Polar Lipids, Alabaster, AL, USA) for 24 h in serum-free medium as previously described (*Pidkovka et al., 2007*) before lysis using the Passive Lysis Buffer (Promega, Madison, WI, USA). NF-κB promoter activity was determined using the Dual-Luciferase Reporter Assay System (Cat# E1910; Promega) in a Turner Biosystems Luminometer model Glomax 20/20 (Mountain View, CA, USA), and normalized to the Renilla luciferase activity of the kit's internal control. Promoter activity was expressed as folds of control activity.

## Enzyme-linked immunosorbent assay (ELISA)

The levels of cellular NF-κB p65 and phosphorylated protein (p-p65) were measured in SMC treated with POVPC as described above. Cells were lysed using the 1X Cell Extraction Buffer PTR provided in the ELISA kit (Abcam, Cambridge, MA, USA). The ELISA was performed using the NF-κB p65 (pS536 + Total) SimpleStep Kit (Abcam) following the manufacturer's protocol. Protein levels were measured using an endpoint reading at OD 455 nm in an Ultramark Microplate Reader (BioRad).

## Statistics

Results are presented as mean ± standard deviation. A two-tailed student $t$-test was used to compare the difference between two groups, and one-way ANOVA followed by a Newman-Keuls test was applied to compare the difference among multiple groups. A $p$ value $<0.05$ was considered significant.

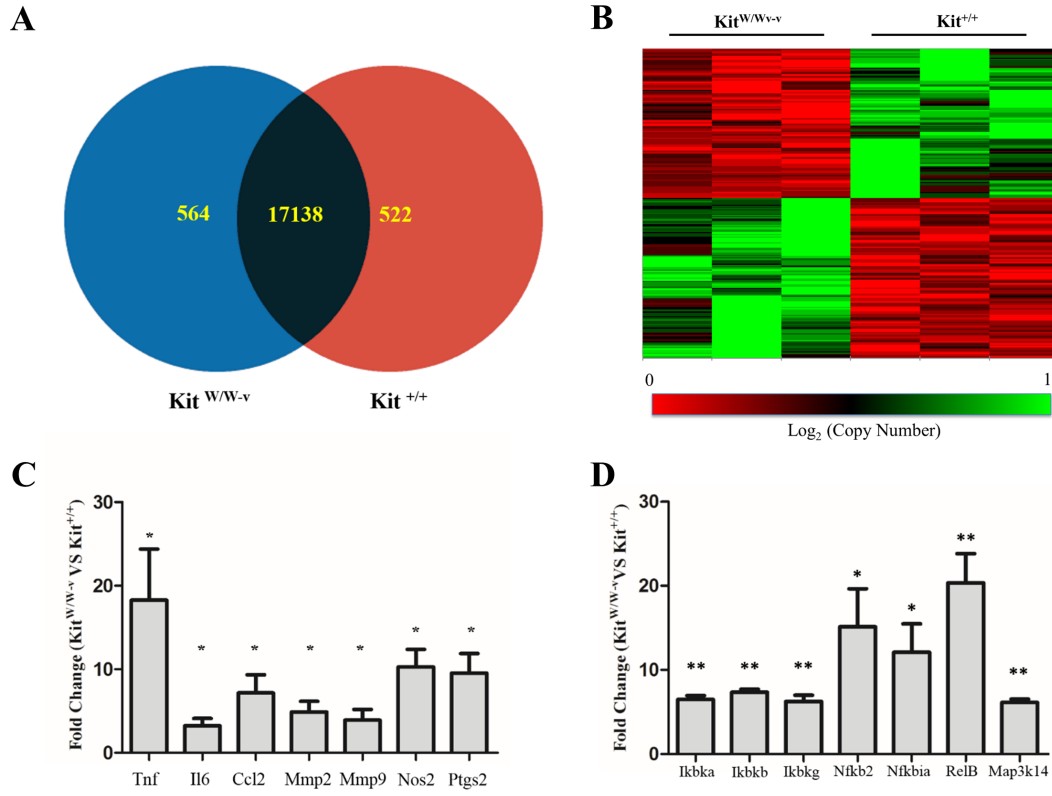

**Figure 1** **Loss of c-Kit function accounts for significant gene expression differences between c-Kit deficient and wild type smooth muscle cells (SMC).** (A) Venn diagram indicating the numbers of differentially up-regulated genes in primary SMC isolated from c-Kit deficient (blue; Kit[W/W−v]) and littermate control mice (red; Kit[+/+]) as determined by microarray analysis. The group of genes in the interception (black area) did not show statistically significant differences by $t$-test between the two strains ($n = 3$ per group). (B) Heat map of differentially expressed genes in primary SMC from c-Kit deficient and littermate control mice. (C) Expression of NF-κB related genes in c-Kit deficient vs. control SMC as determined by real-time PCR. Values are shown as fold change over expression in Kit[+/+] cells; *$p < 0.05$ and **$p < 0.01$ using a two-tailed $t$-test assuming unequal variance, $n = 3$ per group.

# RESULTS

## Different gene expression profiles in c-Kit positive and deficient smooth muscle cells

Considering the reported contribution of c-Kit to vascular remodeling processes (*Hollenbeck et al., 2004*; *Wang et al., 2006*; *Wang et al., 2007*; *Skartsis et al., 2014*; *Young et al., 2016*), we sought for differentially expressed genes between primary SMC isolated from c-Kit deficient (Kit[W/W−v]) and control littermate (Kit[+/+]) mice ($n = 3$ per strain). Out of a total of 34,265 mouse probes queried by microarray, 18,224 yielded a detectable signal above threshold and 1,086 genes were differentially expressed between SMC from both experimental groups ($p < 0.05$) (Figs. 1A–1B). Specifically, 564 and 522 transcripts were significantly up- and down-regulated, respectively, with the loss of c-Kit activity with respect to control SMC (Fig. 1A). No statistically significant differences in expression were detected by microarray in the remaining 17,138 genes.

**Table 1** List of select differentially expressed genes in c-Kit deficient vs. wild type smooth muscle cells.

| Gene symbol | Gene product | Fold change[a] | P-value |
|---|---|---|---|
| **Transcription factors** | | | |
| Crebbp | CREB binding protein | −1.18 | 0.010 |
| Foxo1 | Forkhead box O1 | −2.05 | 0.007 |
| Ilf2 | Interleukin enhancer binding factor 2 | 1.38 | 0.034 |
| Irf3 | Interferon regulatory factor 3 | −1.14 | 0.029 |
| Nfatc1 | Nuclear factor of activated T-cells 1 | −1.36 | 0.040 |
| Nfatc2 | Nuclear factor of activated T-cells 2 | 2.01 | 0.033 |
| Nfatc4 | Nuclear factor of activated T-cells 4 | −2.32 | 0.009 |
| **Cell Adhesion Proteins** | | | |
| Cdh5 | Cadherin 5 | -5.00 | 0.043 |
| Itga9 | Integrin subunit alpha 9 | −2.91 | 0.047 |
| Itga11 | Integrin subunit alpha 11 | −13.28 | 0.034 |
| Pcdh7 | Protocadherin 7 | −2.71 | 0.015 |
| Pcdha1 | Protocadherin alpha 1 | −1.44 | 0.042 |
| Pcdha8 | Protocadherin alpha 8 | 2.26 | 0.002 |
| Selplg | P-selectin glycoprotein ligand 1 | −3.94 | 0.006 |
| **Cytokines/growth factors** | | | |
| Ccl6 | Chemokine (C-C motif) ligand 6 | −8.76 | 0.005 |
| Gdf6 | Growth differentiation factor 6 | −4.10 | 0.019 |
| Ifna14 | Interferon alpha 14 | 1.27 | 0.043 |
| Igf1 | Insulin-like growth factor 1 | −5.06 | 0.027 |
| Pdgfb | Platelet-derived growth factor subunit B | −2.59 | 0.011 |
| Pgf | Placental growth factor | −4.96 | 0.037 |
| Tnfsf9 | Tumor necrosis factor ligand superfamily member 9 | 2.97 | 0.048 |
| **Enzymes** | | | |
| Bmp1 | Bone morphogenetic protein 1 | −2.36 | 0.016 |
| Casp3 | Caspase 3 | 2.06 | 0.040 |
| Ccnd1 | Cyclin D1 | 2.20 | 0.043 |
| Gucy1b3 | Guanylate cyclase 1 soluble subunit beta | −8.95 | 0.033 |
| Ikbkb | Inhibitor of nuclear factor kappa-B kinase subunit beta | 1.34 | 0.001 |
| Lpl | Lipoprotein lipase | −14.24 | 0.022 |
| Mmp23 | Matrix metallopeptidase 23 | −6.61 | 0.049 |
| Pde1a | Ca2+/calmodulin dependent phosphodiesterase 1A | −4.73 | 0.028 |
| Pde2a | Phosphodiesterase 2A | −1.91 | 0.048 |
| Prkg1 | cGMP-dependent protein kinase 1 (PKG) | −9.09 | 0.034 |
| Ptgs1 | Prostaglandin-endoperoxide synthase 1 (COX-1) | 2.22 | 0.027 |
| Sirt1 | Sirtuin 1 | −1.52 | 0.034 |
| Tnfaip3 | TNF alpha induced protein 3 | 3.46 | <0.001 |
| **Receptors** | | | |
| Adra2a | Adrenoceptor alpha 2A | −6.49 | 0.002 |
| Agtr1b | Angiotensin II type 1b receptor | −9.88 | 0.043 |
| Avpr1a | Arginine vasopressin receptor 1A | −6.62 | 0.010 |

**Table 1** (*continued*)

| Gene symbol | Gene product | Fold change[a] | *P*-value |
|---|---|---|---|
| Cxcr4 | Chemokine (C-X-C motif) receptor 4 | −6.55 | 0.040 |
| Igf2r | Insulin like growth factor 2 receptor | −1.48 | 0.012 |
| Il3ra | Interleukin 3 receptor subunit alpha | −2.38 | 0.023 |
| Il20ra | Interleukin 20 receptor alpha | −3.52 | 0.003 |
| Pdgfrb | Platelet-derived growth factor receptor beta | −2.89 | 0.016 |
| Pth1r | Parathyroid hormone 1 receptor | −5.42 | 0.002 |

**Notes.**
[a]Average fold gene expression change in c-Kit deficient smooth muscle cells compared to wild type cells.

Table 1 presents select differentially expressed genes in c-Kit deficient SMC that are relevant for inflammation such as *Ilf2*, *Ifna14*, and *Tnfsf9* (*Zhao et al., 2005*; *Chan et al., 2006*; *Croft, 2009*). We also show decreased expression of the anti-inflammatory genes *Foxo1*, *Gdf6*, *Igf1*, *Igf2r*, and *Lpl* (*Ziouzenkova et al., 2003*; *Sukhanov et al., 2007*; *Savai et al., 2014*; *Hisamatsu et al., 2016*). Lipoprotein lipase (*Lpl*), for example, is 14-fold lower in c-Kit deficient cells than in those isolated from littermate controls. Additional changes in c-Kit deficient SMC are associated with a down-regulation of the contractile SMC phenotype (increased *Tnfaip3* and reduced *Sirt1*) (*Damrauer et al., 2010*; *Huang et al., 2015*) and higher susceptibility to calcification (decreased *Foxo1* and *Pth1r*) (*Cheng et al., 2010*; *Deng et al., 2015*). Finally, we found significant expression differences in genes that code for cell adhesion proteins and for receptors and enzymes that regulate vasomotor responses (Table 1).

Confirmatory real-time RT-PCR assays were performed for select inflammation-related genes that showed a trend by microarray analysis. Tumor necrosis factor (*Tnf*), interleukin 6 (*Il6*), C-C motif chemokine ligand 2 (*Ccl2*), metalloproteinases 2 and 9 (*Mmp2*, *Mmp9*), inducible nitric oxide synthase (*Nos2*), and cyclooxygenase 2 (*Ptgs2*) were significantly up-regulated in c-Kit deficient SMC compared to controls (Fig. 1C).

## Predicted activation of NF-κB signaling in c-Kit deficient cells by pathway analysis

*In silico* pathway analysis was used to predict the molecular pathways affected by the loss of c-Kit in SMC. A total of 71 statistically significant pathways were identified ($p < 0.05$) by the software, 42 of which with a biologically relevant function in SMC. These pathways covered cellular processes such as cell survival and apoptosis, inflammation, cell adhesion, nitric oxide signaling, and lipid metabolism (Table 2). Interestingly, 10 independent molecular pathways were associated with NF-κB signaling, and all of them showed either predicted activation of the entire pathway (5/10; z-scores ranging from 0.258 to 1.265) or of the NF-κB branch (5/10) in c-Kit deficient SMC (Table 2).

## Up-regulation of NF-κB pathway genes in c-Kit deficient smooth muscle cells

The NF-κB pathway plays a fundamental role in SMC differentiation, inflammation, and response to stress signals (*Zahradka et al., 2002*; *Ramana, Friedrich & Srivastava, 2004*; *Mehrhof et al., 2005*; *Mack, 2011*). Therefore, we confirmed the up-regulation of

**Table 2  Select canonical pathways with differentially expressed genes in c-Kit deficient vs. wild type smooth muscle cells.**

| Pathway | Biological function | P-value | Z-score[a] | Predicted status[a] | Differentially expressed genes |
|---|---|---|---|---|---|
| PTEN signaling[b] | Proliferation, apoptosis, de-differentiation, cell migration, inflammation | <0.001 | 0.258 | Activation | Akt2, Casp3, Rac1, Ccnd1, Igf2r, Rac3, Ddr1, Shc1, Ikbkb, Inpp5f, Foxo1, Bmpr1a, Magi2, Magi3, Pdgfrb |
| Death receptor signaling[b] | Apoptosis | 0.003 | 1.265 | Activation | Map2k4, Gas2, Rock1, Diablo, Ikbkb, Casp3, Htra2, Tbk1, Parp1, Birc2 |
| TNFR2 signaling[b] | Cell survival, inflammation | 0.005 | 1.000 | Activation | Map2k4, Ikbkb, Tnfaip3, Tbk1, Birc2 |
| Wnt/ β-catenin signaling | Proliferation, cell survival, cell migration | 0.008 | 0.577 | Activation | Sfrp4, Akt2, Crebbp, Csnk1a1, Fzd9, Ccnd1, Rarg, Fzd8, Cdh5, Dkk3, Sox18, Ppp2r5e, Sfrp1, Wnt5b |
| IRF activation pathway[b] | Inflammation | 0.011 | 1.134 | Activation | Map2k4, Ikbkb, Crebbp, Tbk1, Ifna14, Irf3, Atf2 |
| ERK/MAPK signaling | Proliferation, cell migration, vasoconstriction | 0.028 | 1.069 | Activation | Crebbp, Rac1, Ppp1r14a, Mknk2, Rac3, Nfatc1, Atf2, Pla2g4e, Shc1, Pla2g6, Prkar2b, Prkag2, Rps6ka1, Ppp2r5e |
| TNFR1 signaling[b] | Cell survival, inflammation | 0.044 | 1.000 | Activation | Map2k4, Ikbkb, Casp3, Tnfaip3, Birc2 |
| Wnt/Ca$^{2+}$ pathway[b] | Proliferation, cell migration | <0.001 | −1.000 | Inhibition | Fzd8, Plcb4, Crebbp, Nfatc2, Fzd9, Nfatc4, Wnt5b, Nfatc1, Atf2 |
| AMPK signaling[b] | Cellular senescence, anti-inflammatory, differentiation, vasoconstriction | <0.001 | −0.535 | Inhibition | Pbrm1, Akt2, Crebbp, Ccnd1, Slc2a4, Elavl1, Atf2, Ak6, Prkar2b, Foxo1, Adra2a, Ppm1b, Sirt1, Prkag2, Ppm1a, Ppp2r5e, Ppat, Camkk2 |
| Apoptosis signaling[b] | Apoptosis | <0.001 | −0.302 | Inhibition | Map2k4, Gas2, Rock1, Diablo, Ikbkb, Casp3, Htra2, Rps6ka1, Bcl2a1, Parp1, Birc2 |
| Phospholipase C signaling | Vasoconstriction, stress responses | 0.001 | −0.378 | Inhibition | Rala, Arhgef12, Pld3, Fcgr2a, Arhgef15, Crebbp, Rac1, Ppp1r14a, Nfatc4, Fcgr2b, Rhoh, Nfatc1, Atf2, Pla2g6, Shc1, Pla2g4e, Plcb4, Itpr3, Fcer1g, Nfatc2 |
| Nitric oxide/GC signaling | Vasodilation | 0.005 | −0.302 | Inhibition | Bdkrb2, Kng1, Pde2a, Akt2, Prkg1, Prkar2b, Itpr3, Prkag2, Pde1a, Gucy1b3, Pgf |
| Integrin signaling | Cell adhesion, cell migration, proliferation, apoptosis, stress responses, differentiation | 0.029 | −1.387 | Inhibition | Map2k4, Akt2, Rala, Rac1, Rac3, Rhoh, Pdgfb, Rock1, Arhgap5, Shc1, Itga11, Itga9, Actn4, Tspan6, Nedd9 |
| Adipogenesis pathway | Lipid synthesis and storage | <0.001 | N.D. | Could not be predicted | Nr2f2, Sin3b, Fzd9, Nfatc4, Rbp1, Slc2a4, Fzd8, Cdk5, Foxo1, Bmpr1a, Lpl, Sirt1, Ctbp2, Clock, Fabp4, Rps6ka1, Stat5b |

**Table 2** (*continued*)

| Pathway | Biological function | *P*-value | *Z*-score[a] | Predicted status[a] | Differentially expressed genes |
|---|---|---|---|---|---|
| Fibroblast inflammatory pathway[b] | Proliferation, cell migration, differentiation, inflammation | 0.012 | N.D. | Could not be predicted | Map2k4, Sfrp4, Akt2, Crebbp, Csnk1a1, Rac1, Fzd9, Nfatc4, Ccnd1, Nfatc1, Pdgfb, Pgf, Atf2, Rock1, Ikbkb, Fzd8, Plcb4, Dkk3, Nfatc2, Sfrp1, Wnt5b |
| Gαq signalingy[b] | Proliferation, cell migration, vasoconstriction | 0.026 | 0.000 | Could not be predicted | Rock1, Ikbkb, Plcb4, Akt2, Pld3, Agtr1b, Itpr3, Nfatc2, Nfatc4, Avpr1a, Rhoh, Nfatc1 |

**Notes.**

[a]*Z*-score and predicted functional status in c-Kit deficient smooth muscle cells compared to wild type cells. The *z*-score measures how close the gene expression data matches the experimentally observed direction of pathway regulation in the literature. A positive *z*-score predicts activation, while a negative *z*-score indicates inhibition. N.D., could not be determined.

[b]NF-κB associated signaling pathway.

components of this pathway in c-Kit deficient cells by real-time RT-PCR (Fig. 1D). We found significantly higher expression levels of genes that are part of both the canonical (*Ikbka, Ikbkb, Ikbkg, Nfkbia*) and alternative (*Ikbka, Map3k14, Nfkb2, RelB*) NF-κB signaling pathways in Kit$^{W/W-v}$ vs. Kit$^{+/+}$ SMC.

### Increased activity of the canonical NF-κB pathway in stimulated c-Kit deficient cells

Given that the inhibitor of the canonical NF-κB pathway (*Nfkbia*) and the negative regulator *Tnfaip3* were up-regulated in c-Kit deficient SMC (Table 1 and Fig. 1D), we turned to demonstrate the relationship between c-Kit expression and functional activity of the NF-κB signaling pathway. To further validate our findings, we rescued c-Kit expression in Kit$^{W/W-v}$ SMC by lentiviral transduction (Fig. S1).

POVPC-stimulated SMC with deficient c-Kit expression showed higher NF-κB transcriptional activity compared to wild type and c-Kit rescued cells as determined by a dual luciferase reporter assay (Fig. 2A). Accordingly, a significantly higher ratio of the S536-phosphorylated/total p65 factor was detected in Kit$^{W/W-v}$ SMC vs. wild type and rescued cells (Fig. 2B), demonstrating increased availability of active p65 in c-Kit deficient SMC for nuclear translocation and promoter binding (*Lawrence, 2009*). Finally, we evaluated the protein concentrations of the pro-inflammatory mediators MMP-2 and MCP-1 in POVPC-stimulated SMC, two factors that are regulated by NF-κB (*Lee et al., 2008*; *Song et al., 2016a*). In agreement with the enhanced transcriptional activity shown above, the protein expressions of both MMP-2 and MCP-1 were significantly higher in Kit$^{W/W-v}$ SMC compared to wild type and rescued cells (Fig. 2C). To control for off-target effects of POVPC stimulation on c-Kit expression, we demonstrated that this treatment did not modify the cellular levels of neither c-Kit nor its ligand SCF (Fig. S2).

### c-Kit inhibits NF-κB activity through TAK1/NLK in smooth muscle cells

Previous studies indicate an association between c-Kit, Lyn (a member of the Src family of non-receptor tyrosine kinases), and TAK1 (*Drube et al., 2015*), a negative regulator of NF-κB signaling (*Ajibade et al., 2012*). Therefore, we assessed whether this latter factor

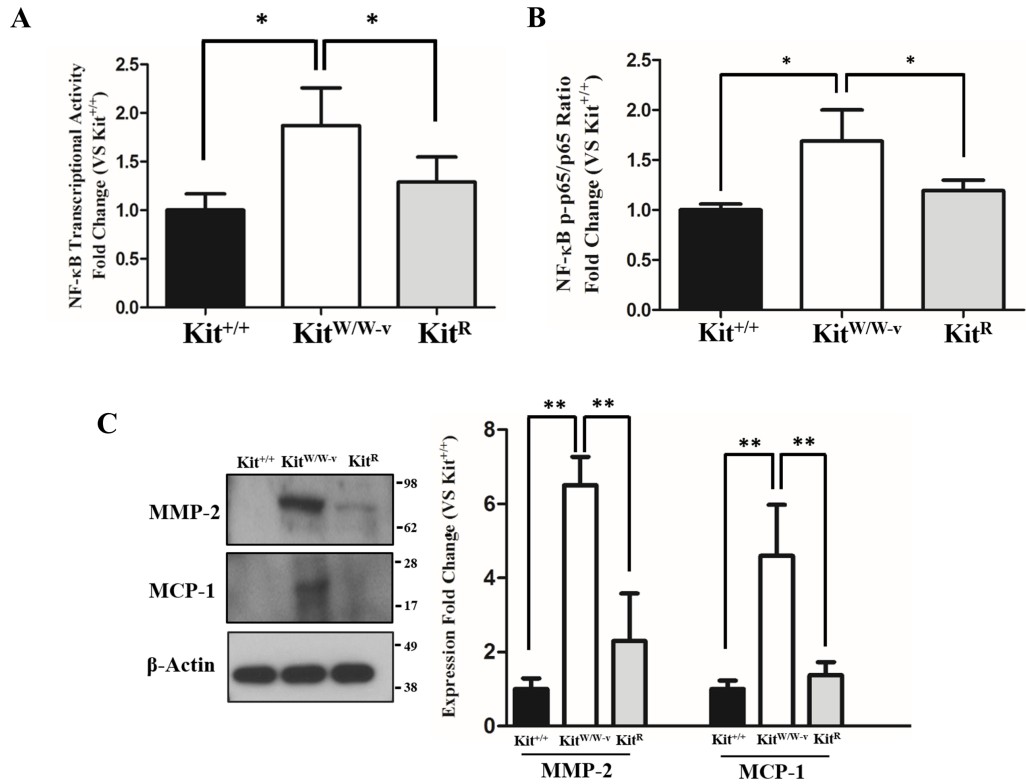

**Figure 2  Loss of c-Kit function in primary smooth muscle cells (SMC) is associated with increased NF-κB activity.** (A) NF-κB transcriptional activity in c-Kit deficient (Kit$^{W/W-v}$), control (Kit$^{+/+}$), and c-Kit rescued SMC (Kit$^R$) after 24-hour treatment with POVPC (50 μg/ml), as determined by dual-luciferase reporter assay. Transcriptional activity is represented as the mean ± standard deviation (SD) of the Firefly/Renilla luciferase ratio normalized with respect to the control group (Kit$^{+/+}$) ($n = 3$ independent experiments). (B) Phosphorylated (pS536) protein levels of the NF-κB p65 subunit in POVPC-treated c-Kit deficient, control, and c-Kit rescued SMC as determined by ELISA. Values are expressed as the mean ± SD of the p-p65/total p65 ratio normalized with respect to the control group (Kit$^{+/+}$) ($n = 3$ independent experiments). (C) Protein expression of the NF-κB related pro-inflammatory mediators MMP-2 and MCP-1 in POVPC-treated c-Kit deficient, control, and c-Kit rescued SMC as determined by Western blot. Molecular weight markers are shown on the right side of the gel. Protein expression is expressed as the mean ± SD of the MMP-2/β-actin and MCP-1/β-actin signal ratios normalized with respect to the control group (Kit$^{+/+}$) ($n = 3$ per cell type). *$p < 0.05$ and **$p < 0.01$ using a one-way ANOVA followed by a Newman-Keuls test.

or its downstream partner NLK (*Yasuda et al., 2004*; *Li et al., 2014*) were responsible for the observed inhibition of the NF-κB pathway in c-Kit expressing SMC. We found that the protein expressions of both TAK1 and NLK were reduced or lost in Kit$^{W/W-v}$ SMC compared to wild type or c-Kit rescued cells (Fig. 3A). Next, we selectively knocked down TAK1 or NLK in Kit$^{+/+}$ SMC (Figs. 3B–3C), and showed that this genetic manipulation restored the NF-κB transcriptional activity, phosphorylated/total p65 ratio, and protein expressions of MMP-2 and MCP1 in POVPC-stimulated c-Kit wild type SMC (Figs. 3D–3F). Lastly, we demonstrated by co-IP a physical interaction between all c-Kit, Src, TAK1, and NLK (Fig. 4), further supporting a direct relationship in NF-κB regulation. Altogether, these experiments demonstrate that c-Kit inhibits NF-κB signaling in SMC through the actions of TAK1 and NLK.

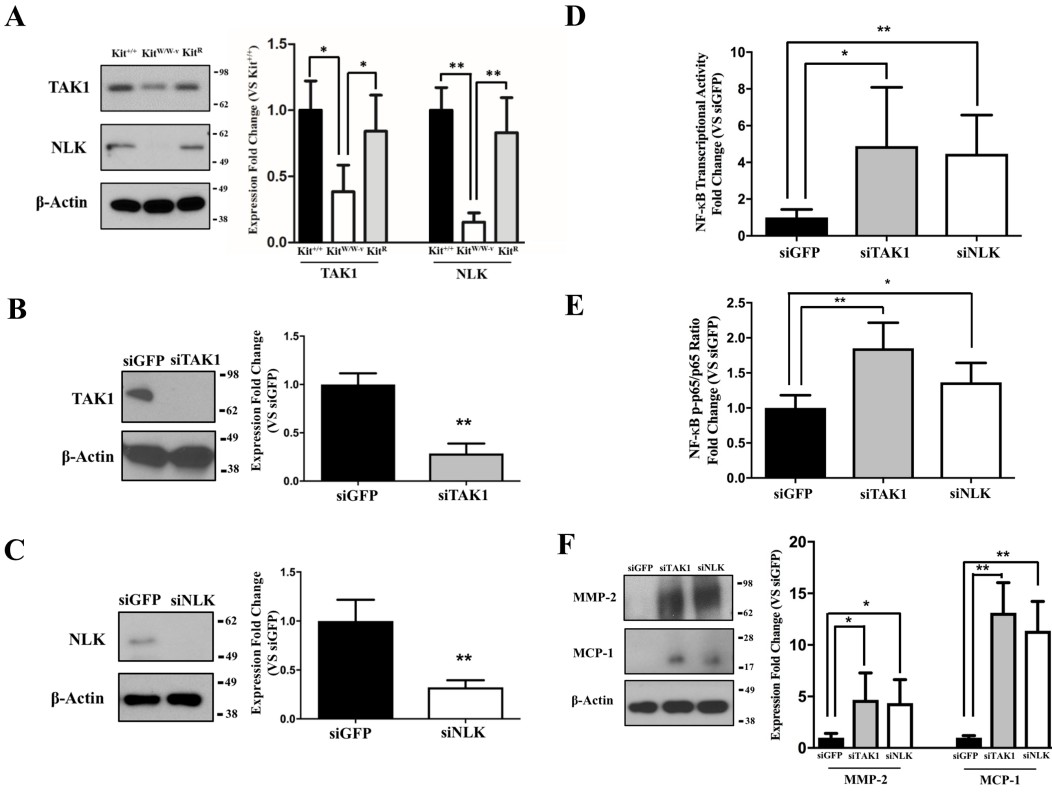

**Figure 3  c-Kit inhibits NF-κB activity in smooth muscle cells (SMC) through the actions of TAK1 and NLK.** (A) Protein expression of the TAK1 and NLK regulatory proteins in c-Kit deficient (Kit[W/W−v]), control (Kit[+/+]), and c-Kit rescued SMC (Kit[R]) as determined by Western blot. Protein expression is expressed as the mean ± standard deviation (SD) of the TAK1/$\beta$-actin and NLK/$\beta$-actin signal ratios normalized with respect to the control group (Kit[+/+]) ($n = 3$ per cell type). (B–C) Protein expression of TAK1 (B) and NLK (C) in Kit[+/+] cells transduced with lentivirus-encoded siRNAs of the corresponding targets or GFP control. Protein expression is expressed as the mean ± SD of the TAK1/$\beta$-actin and NLK/$\beta$-actin signal ratios normalized with respect to the siGFP-treated group ($n = 3$ independent experiments). (D) NF-κB transcriptional activity in Kit[+/+] SMC transduced with lentivirus-encoded siRNAs complementary to TAK1, NLK, or GFP after 24-hour treatment with POVPC (50 μg/ml), as determined by dual-luciferase assay. Transcriptional activity is represented as the mean ± SD of the Firefly/Renilla luciferase ratio normalized with respect to the siGFP-treated group ($n = 3$ independent experiments). (E) Phosphorylated (pS536) protein levels of NF-κB p65 in POVPC-treated Kit[+/+] SMC transduced with lentivirus-encoded siRNAs as determined by ELISA. Values are expressed as the mean ± SD of the p-p65/total p65 ratio normalized with respect to the siGFP-treated group ($n = 3$ independent experiments). (F) Protein expression of the pro-inflammatory mediators MMP-2 and MCP-1 in POVPC-treated Kit[+/+] SMC transduced with lentivirus-encoded siRNAs as determined by Western blot. Protein expression is expressed as the mean ± SD of the MMP-2/$\beta$-actin and MCP-1/$\beta$-actin signal ratios normalized with respect to the siGFP-treated group ($n = 3$ independent experiments). Molecular weight markers are shown on the right side of the gels. *$p < 0.05$ and **$p < 0.01$ using a one-way ANOVA followed by a Newman-Keuls test.

## DISCUSSION

Vascular SMC are characterized by tremendous phenotypic diversity (*Yoshida & Owens, 2005*). Moreover, their contribution to vascular pathogenesis greatly depends on their phenotype and state of differentiation (*Archer, 1996*; *Yoshida & Owens, 2005*). Expression of the c-Kit receptor in SMC has been associated with various vascular pathologies in

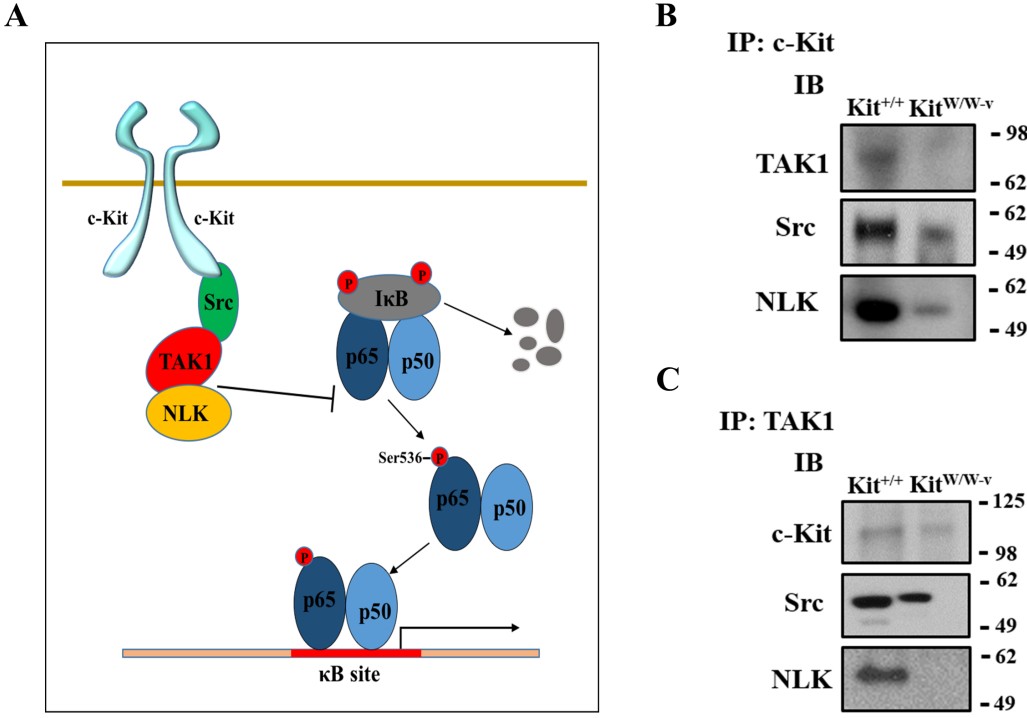

**Figure 4** **c-Kit forms a molecular complex with the regulatory proteins TAK1, Src, and NLK in smooth muscle cells (SMC).** (A) Diagram illustrating the proposed molecular complex between c-Kit, TAK1, Src, and NLK in SMC and their inhibitory function on NF-κB transcriptional activity. (B–C) Co-immunoprecipitation experiments in control (Kit$^{+/+}$) and c-Kit deficient (Kit$^{W/W-v}$) SMC using anti-c-Kit (B) and anti-TAK1 antibodies (C) to pull down protein complexes. Molecular weight markers are shown on the right side of the gels, while antibodies used to detect eluted proteins are indicated on the left. IP, immunoprecipitation; IB, immunoblot.

both animal models (*Wang et al., 2006*; *Wang et al., 2007*; *Skartsis et al., 2014*; *Young et al., 2016*) and human samples (*Hollenbeck et al., 2004*; *Skartsis et al., 2014*). In contrast, it has also proven protective in models of atherosclerosis (*Song et al., 2016b*). In light of this evidence, there is little information on how c-Kit influences the phenotypes of SMC or the mechanisms by which they contribute to pathology. Our work reveals that the absence of c-Kit modified the expression of approximately 6% of the genes that were detected by microarray in SMC from c-Kit mutant and littermate control mice. Furthermore, we provide evidence that c-Kit suppresses NF-κB signaling in SMC and decreases the production of pro-inflammatory mediators under stimulus.

Using *in silico* pathway analysis, we first demonstrated that c-Kit signaling influences a wide variety of cellular processes in SMC. Specifically, we found evidence of a pro-synthetic and pro-inflammatory phenotype in SMC secondary to the loss of this receptor. This is particularly evident by the potential dysregulation of lipid metabolism as indicated by a 14-fold decrease in *Lpl* gene expression. While increased vascular lipoprotein lipase can be pro-atherogenic (*Clee et al., 2000*), it is also believed to have anti-inflammatory properties both by generating metabolic PPAR agonists and inhibiting NF-κB activity (*Ziouzenkova et al., 2003*; *Kota et al., 2005*). c-Kit deficient cells also have decreased expression of the
anti-inflammatory and anti-atherogenic factor IGF-1 (*Sukhanov et al., 2007*) and increased susceptibility to calcification due to the down-regulation of the *Foxo1* and *Pth1r* genes (*Cheng et al., 2010*; *Deng et al., 2015*). It is possible that these changes explain the increased severity of atherosclerosis in c-Kit mutant animals (*Song et al., 2016b*). In addition, c-Kit deficient SMC appear to respond differently to vasomotor stimuli. Their gene expression profile indicates a significantly lower expression of vasoconstrictive G-protein coupled receptors such as the angiotensin II receptor type 1B and the arginine vasopressin receptor 1A. The response to nitric oxide may be also impaired in these cells due to a lower expression of guanylate cyclase 1 soluble subunit beta and cGMP-dependent protein-kinase type I. In the absence of functional experiments, it is not clear what is the biological impact of the above differences in c-Kit deficient SMC compared to their wild type counterparts. However, these observations warrant further investigations.

Interestingly, 24% of the differentially regulated pathways identified were associated with NF-κB signaling. Furthermore, both the *in silico* analysis and our experimental data demonstrated activation of this pathway in c-Kit deficient SMC with respect to those from littermate controls. NF-κB signaling is critical for the regulation of proliferation, differentiation, stress responses, and inflammatory processes in vascular SMC (*Zahradka et al., 2002*; *Ramana, Friedrich & Srivastava, 2004*; *Mehrhof et al., 2005*; *Mack, 2011*). Whether NF-κB activation is associated with increased proliferation or apoptosis in SMC is dependent on the upstream stimuli and the type of vessel (*Zahradka et al., 2002*; *Mehrhof et al., 2005*; *Ogbozor et al., 2015*). A recent study demonstrated that NF-κB activation led to increased proliferation in fibroblasts, while inducing apoptosis and inflammation in SMC (*Mehrhof et al., 2005*). On the other hand, NF-κB was shown to be an important intracellular mediator of angiotensin II responses, leading to SMC proliferation and migration under these conditions (*Zahradka et al., 2002*). In terms of cell differentiation, NF-κB is known to repress myocardin activity and cause down-regulation of SMC contractile genes (*Tang et al., 2008*). This molecular interaction has been implicated in the origin of synthetic SMC under inflammatory processes such as atherosclerosis (*Mack, 2011*). Interestingly, the reduced expression of *Sirt* and increased mRNA level of *Tnfaip3* in c-Kit deficient cells are independently associated with down-regulation of contractile genes in SMC (*Damrauer et al., 2010*; *Huang et al., 2015*). These observations are in agreement with the predicted de-differentiated phenotype of c-Kit deficient SMC (*Davis et al., 2009*) and with the reported atheroprotective role of the c-Kit receptor (*Song et al., 2016b*). TNFAIP3 normally provides a negative regulatory loop for the NF-κB pathway, including the decreased downstream production of the MCP-1 inflammatory mediator (*Patel et al., 2006*; *Giordano et al., 2014*). Down-regulation of the *Crebbp* transcription factor is also thought to reduce NF-κB transcriptional activity (*Yang et al., 2010*). Nonetheless, neither higher *Tnfaip3* expression nor less *Crebbp* in c-Kit deficient SMC seem to have an appreciable inhibitory effect on NF-κB signaling, as demonstrated by our functional experiments and the increased protein expressions of the MMP-2 and MCP-1 factors.

Typical stimuli for NF-κB activation include cytokines, endotoxins, lipids, and mechanical stress (*Maziere et al., 1996*; *De Martin et al., 2000*; *Kumar & Boriek, 2003*). For example, the oxidized phospholipid POVPC has been previously used to induce

inflammation in vascular SMC (*Pidkovka et al., 2007*; *Lu et al., 2013*) and NF-κB activation (*Pegorier et al., 2006*; *Vladykovskaya et al., 2011*; *Lu et al., 2013*). As predicted by the *in silico* analysis, c-Kit deficiency in SMC led to higher levels of NF-κB transcriptional activity, phosphorylation of its key subunit p65, and expression of the NF-κB regulated inflammatory mediators MMP-2 and MCP-1 under POVPC challenge. Gene members of the non-canonical NF-κB pathway and other inflammatory mediators were also up-regulated in mutant SMC.

The role of c-Kit as a negative regulator of the NF-κB pathway and related inflammation has been previously observed in other cell types and under different stimuli (*Jin et al., 2013*; *Micheva-Viteva et al., 2013*). Pharmacological inhibition of c-Kit results in increased activation of NF-κB in HEK293 cells and secretion of TNFα in dendritic cells and the THP-1 monocytic cell line in response to bacterial infection (*Micheva-Viteva et al., 2013*). Lower expressions of SCF and c-Kit were also associated with increased NF-κB signaling and oxidative stress in gastric smooth muscle (*Jin et al., 2013*).

Our experiments further revealed that c-Kit reduces NF-κB mediated inflammation via a direct molecular interaction with the NF-κB negative regulators TAK1 and NLK (*Yasuda et al., 2004*; *Ajibade et al., 2012*; *Li et al., 2014*). The physical association between c-Kit, Lyn (a member of the Src family of non-receptor tyrosine kinases), and TAK1 has been previously observed in the HEK293T cell line, where these proteins form a signalosome that interacts with IKKβ, one of the catalytic units of the IκB kinase (IKK) complex (*Drube et al., 2015*). Nonetheless, the inhibitory activity of TAK1 on the NF-κB signaling pathway appears to be cell-specific, since in some cells it can be activating (*Israel, 2010*; *Ajibade et al., 2012*). In the inhibitory instances, TAK1 blocks the phosphorylation and inactivates IKK (*Ajibade et al., 2012*), which in turn is unable to phosphorylate and induce proteosomal degradation of the IκB inhibitors of the NF-κB pathway (*Karin, 1999*; *Israel, 2010*). When active, IκB proteins prevent the nuclear translocation of p65/RelA complexes (*Karin, 1999*; *Israel, 2010*). NLK also functions as an inhibitor of IKK phosphorylation (even in cells where TAK1 acts as an activator) (*Li et al., 2014*). Therefore, our data indicate that in SMC the roles of TAK1 and NLK may be redundant.

In conclusions, our study demonstrates that c-Kit expression in SMC has an anti-inflammatory role. Our mechanistic studies contradict the existing belief about the noxious effect of SCF/c-Kit signaling to the vasculature. It is noteworthy to recognize that such idea originated from descriptive studies and models of post injury IH. The current knowledge describes the expression of SCF and its receptor c-Kit in endothelial cells and SMC (*Hollenbeck et al., 2004*; *Matsui et al., 2004*; *Wang et al., 2007*; *Skartsis et al., 2014*), and suggests a key role for this signaling pathway in myofibroblast mobilization towards the neointima (*Hollenbeck et al., 2004*; *Skartsis et al., 2014*). Increased survival of SCF-treated SMC through Akt has also been demonstrated (*Wang et al., 2007*). In contrast, one recent study revealed that SCF/c-Kit signaling protects hyperlipidemic ApoE$^{-/-}$ mice from excessive atherosclerotic plaque deposition (*Song et al., 2016b*). This apparent discrepancy may reflect the existing differences between IH (restenosis) and atherosclerosis in terms of etiology, natural history, culprit lesions, and progenitor cell contribution to disease progression. Therefore, our results suggest that while c-Kit positive cells have a

detrimental effect on proliferative vascular lesions, their presence may prove protective in inflammatory conditions such as atherosclerosis. In addition, we propose a novel pathway for NF-κB regulation downstream of c-Kit activation. This information could be relevant in the setting of atherosclerosis disease development and complications, and may shed light on new proliferation control mechanisms to address IH after vascular injury.

### Funding

This work was supported by the National Institutes of Health grant R01-HL-109582 to RIVP. The funders had no role in study design, data collection and analysis, decision to publish, or preparation of the manuscript.

### Grant Disclosures

The following grant information was disclosed by the authors:
National Institutes of Health: R01-HL-109582.

### Competing Interests

The authors declare that there are no competing interests.

### Author Contributions

- Lei Song conceived and designed the experiments, performed the experiments, analyzed the data, wrote the paper, prepared figures and/or tables, reviewed drafts of the paper.
- Laisel Martinez conceived and designed the experiments, analyzed the data, wrote the paper, prepared figures and/or tables, reviewed drafts of the paper.
- Zachary M. Zigmond performed the experiments, reviewed drafts of the paper.
- Diana R. Hernandez and Roberta M. Lassance-Soares analyzed the data, reviewed drafts of the paper.
- Guillermo Selman conceived and designed the experiments, performed the experiments, analyzed the data, reviewed drafts of the paper.
- Roberto I. Vazquez-Padron conceived and designed the experiments, analyzed the data, contributed reagents/materials/analysis tools, wrote the paper, prepared figures and/or tables, reviewed drafts of the paper.

### Animal Ethics

The following information was supplied relating to ethical approvals (i.e., approving body and any reference numbers):

All animal procedures were performed according to the National Institutes of Health guidelines (Guide for the Care and Use of Laboratory Animals) and approved by the University of Miami Miller School of Medicine Institutional Animal Care and Use Committee (protocol 15-114).

### Patent Disclosures

The following patent dependencies were disclosed by the authors:

RNA purification method (US Patent 7049102; Inventors Russell N. Van Gelder, Mark E. Von Zastrow, Jack D. Barchas, James H. Eberwine; Filing date Nov 15, 2000).

## Data Availability

The raw data has been supplied as a Supplementary File.

## Supplemental Information

Supplemental information for this article can be found online at http://dx.doi.org/10.7717/peerj.3418#supplemental-information.

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
