# Peer review of "c-Kit modifies the inflammatory status of smooth muscle cells"

_PeerJ, doi:10.7717/peerj.3418_

## Round 0.1 · original submission · Major Revisions

· Academic Editor

Major Revisions

As you can see, your paper was reviewed by two experts in the field who expressed different opinions about the extent of the corrections to be made. I suggest that you try to take advantage of these suggestions to make your paper stronger. If you have solid arguments to reject some of them, please make them clear both to me and to the reviewers so that we can reach a decision. I cannot give you any guarantee of acceptance at this stage without seeing how you handled the suggestions made. Please, do not hesitate to ask for an extension of the delay for resubmission if you decide to perform additional experiments in order to better meet the reviewers criticisms.

Reviewer 1 ·

Basic reporting

The study is a starightforward to elucidate the function of c-Kit in SMC biology and valuable for further clarification.

Experimental design

Generally, the study is well designed for experimental approach.

Validity of the findings

In the function of c-Kit in SMS biology where SMC such as aortic or vascular endothelial linked SMC are a direct mediator of the atherosclerosis and inflammatory proliferation upon NF-kB, AP-1 and c-fos kinations. The authors have found and suggested a new mechanism by which c-Kit inhibits NF-κB target gene regulation in SMC. For the c-Kit-regulated NF-kB suppression, c-Kit is dependent on the TGFβ-activated kinase 1 (TAK1) and Nemo-like kinase (NLK) activity.
This is an interesting study with a new information of c-Kit-NF-kB-TAK1/NLK pathway-mediated SMC phenotype changes. However, the present manuscript is suffered from the insufficient mechanistic explanation on the global SMC biology. The followings are some of the major criticisms to justify the author's claims.
1. How about the distribution of c-Kit expression in various SMC types ? What is the general function of c-Kit in SMS biology, for example, with regard to the ligands.
2. Upon TNF-a ttreatment or proinflammatory condition, how about the expression level of c-Kit (please note that NF-kB and other related transcription factors are highly expressed in the same condition.
3. For the NF-kB-targeted gene during inflammation-regulated SMC proliferation, MMP-9 gene expression is exclusively subjected rather than MMP-2 due to MMP-9 promoter sequence. Therefore, MMP-9 should be examined and also COX-2 or iNOS...
4. Fig. 4 is wrongly illustrated because pNF-kB binds the promoter region of the target genes.

Comments for the author

The manuscript should be returned back to the authors for fundamentally additional experiments, as raised above.
Although the basic findings of the present study are valuable for further evidences to be accepted in journals. Thus, the present form should be rejected.

·

Basic reporting

The manuscript by Song et al. describes the original finding that loss of c-Kit in smooth muscle cells induces activation of both canonical and alternative nuclear factor kappa B pathways. This is relevant to cardiovascular diseases and other settings where there is dysregulation of smooth muscle cell function.

Experimental design

The authors used microarrays datasets and in vitro signalling to reach their conclusions. Their approach is adequate.

Validity of the findings

The findings are of extreme interest both for vascular biology and pathogenesis of disease. The findings also raise the key question of whether the results are cell type specific. The usage of NFkB signalling has been shown before to be cell specific and often species specific. The contrasting literature the authors refer to could be explained in this context. It would be great if the authors had capacity to perform some in vitro assays to compare their findings in another cell type, for instance an inflammatory cell.

Comments for the author

The findings are novel and valid. It would be useful to know if the signalling effects are cell-type specific.

---

## Round 0.2 · accepted · Accept

· Academic Editor

Accept

We apologize for the delay, but we wished to have the opinion of the original reviewers. As these did not respond to our invitation, I undertook to examine you rebuttal my-self. I was satisfied with the way you handled the remarks and suggestions raised by the reviewers.